# The Cytokine, Chemokine, and Growth Factor Network of Prenatal Depression

**DOI:** 10.3390/brainsci13050727

**Published:** 2023-04-26

**Authors:** Michael Maes, Yoshiko Abe, Wandee Sirichokchatchawan, Junpen Suwimonteerabutr, Ussanee Sangkomkamhangd, Abbas F. Almulla, Sirina Satthapisit

**Affiliations:** 1Department of Psychiatry, Faculty of Medicine, Chulalongkorn University and King Chulalongkorn Memorial Hospital, the Thai Red Cross Society, Bangkok 10330, Thailand; 2Kyung Hee University, 26 Kyungheedae-ro, Dongdaemun-gu, Seoul 02447, Republic of Korea; 3Department of Psychiatry, Medical University of Plovdiv, 4002 Plovdiv, Bulgaria; 4Research Institute, Medical University of Plovdiv, 4002 Plovdiv, Bulgaria; 5IMPACT Strategic Research Center, Barwon Health, Geelong 3220, Australia; 6College of Public Health Sciences (CPHS), Chulalongkorn University, Bangkok 10330, Thailand; 7Department of Public Health, Dokkyo Medical University School of Medicine, Tochigi 321-0293, Japan; 8Health and Social Sciences and Addiction Research Unit, Chulalongkorn University, Bangkok 10330, Thailand; 9Department of Obstetrics, Gynaecology and Reproduction, Faculty of Veterinary Science, Chulalongkorn University, Bangkok 10330, Thailand; 10Department of Obstetrics and Gynecology, Khon Kaen Hospital, Khon Kaen 40000, Thailand; 11Medical Laboratory Technology Department, College of Medical Technology, The Islamic University, Najaf 54001, Iraq; 12Department of Psychiatry, Khon Kaen Hospital, Khon Kaen 40000, Thailand

**Keywords:** major depression, neuro-immune, inflammation, cytokines, affective disorders, psychiatry

## Abstract

Background: Neuro-immune pathways are engaged in antenatal and postpartum depression. Aims: To determine if immune profiles influence the severity of prenatal depression above and beyond the effects of adverse childhood experiences (ACE), premenstrual syndrome (PMS), and current psychological stressors. Methods: Using the Bio-Plex Pro human cytokine 27-plex test kit, we assayed M1 macrophage, T helper (Th)-1, Th-2, Th-17, growth factor, chemokine, and T cell growth immune profiles as well as indicators of the immune inflammatory response system (IRS) and compensatory immunoregulatory system (CIRS) in 120 pregnant females in the early (<16 weeks) and late (>24 weeks) pregnancy. The Edinburgh Postnatal Depression Scale (EPDS) was used to assess severity of antenatal depression. Results: Cluster analyses showed that the combined effects of ACE, relationship dissatisfaction, unwanted pregnancy, PMS, and upregulated M1, Th-1, Th-2, and IRS immune profiles and the ensuing early depressive symptoms shape a stress-immune-depression phenotypic class. Elevated IL-4, IL-6, IL-8, IL-12p70, IL-15, IL-17, and GM-CSF are the cytokines associated with this phenotypic class. All immune profiles (except CIRS) were significantly associated with the early EPDS score, independent of the effects of psychological variables and PMS. There was a shift in immune profiles from early to late pregnancy, with an increase in the IRS/CIRS ratio. The late EPDS score was predicted by the early EPDS score, adverse experiences, and immune profiles, mainly the Th-2 and Th-17 phenotypes. Conclusions: Activated immune phenotypes contribute to early and late perinatal depressive symptoms above and beyond the effects of psychological stressors and PMS.

## 1. Introduction

Antenatal depression (AD) is a worldwide public health problem with negative effects on mother and child [1]. Antenatal depression increases the risk of postnatal depression, low birth weight, operative delivery, preterm birth, preeclampsia, mother and familial distress, and suicide risk, as well as impacting neurocognitive development and behaviors in the children [1,2,3,4,5,6,7].

Psychological stressors, such as relationship unhappiness, unwanted pregnancy, partner abuse, adverse childhood experiences (ACE), and lowered social support greatly predict the severity of antenatal depression [7]. Previous meta-analysis and reviews reported that ACE, such as physical, emotional, and sexual abuse, physical and emotional neglect, and family dysfunctions, substantially enhance the likelihood of antenatal depression [7,8,9,10,11]. Recently, it was established that less social support from friends partially mediates the impact of childhood neglect on prenatal depression severity [7]. In fact, ACE (especially psychological and physical abuse and neglect), emotional or physical abuse by the spouse, unexpected pregnancy, and dissatisfaction with the intimate spouse relationship, predicted a larger part of the variance in severity of antenatal depression as assessed with the Edinburgh Postnatal Depression Scale (EPDS) score [7].

In addition, the severity of premenstrual syndrome (PMS), as measured by the Premenstrual Symptoms Screening Test (PSST), was a substantial predictor of the severity of antenatal depression [12]. PMS or better Menstrual Cycle—Associated Syndrome (MCAS, which offers better diagnostic criteria than PMS) [13,14] is characterized by increased levels of affective symptoms, including depressed mood, tearfulness, mood swings, irritability, and tension, and physiosomatic symptoms, including food cravings, headache, abdominal bloating, and breast pain [13,14]. In women with MCAS/PMS, these symptoms are elevated throughout the menstrual cycle, but deteriorate 6 days and reach a peak 2 days before menstruation begins [13,14,15,16]. In fact, the regression on PSST scores (affective, physiosomatic and inference with daily functioning) coupled with partner abuse and ACE explained a larger part of the variance in the EPDS score, when assessed in the early pregnancy phase (<16 gestational weeks) [7]. The latter study reported that the EPDS score assessed after gestational week 24 was strongly associated with the affective component of the PSST [7].

Neuro-immune and neuro-oxidative stress pathways are engaged in antenatal and postpartum depression [1,17,18]. Prenatal depression is associated with exaggerated pregnancy-specific changes in these pathways, including decreased levels of zinc and increased C-reactive protein, indicating a mild inflammatory response, and increased indices of protein oxidation and nitric oxide metabolites, indicating increased nitro-oxidative stress [1,18]. Elevated levels of IL-1β and IL-6 are related to the severity of depression throughout the prenatal and postnatal periods [19]. Major depressive disorder (MDD) in non-pregnant people is characterized by activated M1 macrophage, T helper (Th)-1, Th-2, Th-17, T cell proliferation, chemokine, and growth factor profiles, as well as the immune-inflammatory responses system (IRS) and the compensatory immunoregulatory system (CIRS) [20,21,22].

Moreover, the same neuro-immune and neuro-oxidative pathways are implicated in the pathophysiology of MCAS/PMS, menstruation distress, and ACE [21,23,24,25]. Increased levels of neurotoxic chemokines such as CCL2, CCL11, CCL5, and CXCL10 predict affective and physiosomatic symptoms during the menstrual cycle, whereas menstruation-related physiosomatic (pain, cramps, gastro-intestinal) symptoms are associated with CCL5 and inflammatory markers [23,24]. Crucially, ACE stimulates the networks of cytokines and growth factors linked with MDD [21]. In addition, in humans, psychosocial stressors may induce mild immune-inflammatory responses [26,27].

Nevertheless, no research has evaluated whether prenatal depression is associated with abnormalities in the M1, Th-1, Th-2, Th-17, T cell proliferation, chemokine, IRS, and CIRS profiles, above and beyond the effects of PMS and adverse experiences. To determine if these immune profiles influence the severity of prenatal depression beyond the effects of ACE and PMS, the present investigation was conducted. The specific hypothesis is that immune profiles, and especially the M1, Th-1, Th-17, and IRS profiles, are associated with antenatal depression, because these three profiles are strongly associated with non-pregnancy-associated MDD [20].

## 2. Methods

### 2.1. Participants

This research included 120 pregnant women at the Antenatal Care (ANC) Center at Khon Kaen Hospital, a provincial hospital located in Khon Kaen, Thailand. Khon Kaen is a province in Thailand’s northeast. Using predetermined inclusion and exclusion criteria, a professional researcher selected the pregnant participants. Pregnant women between the ages of 18 and 49 who resided in the province of Khon Kaen and were able to read and write Thai were eligible. In addition, their pregnancy had not yet reached 16 weeks gestation, and they wanted to get ANC treatment in the hospital until giving birth. Our study is a prospective cohort study whereby we examined two repeated measurements of depression scores and immune data from the early (gestational week < 16) to late (gestational week > 24) prenatal phase. After the first early phase assessment, twenty-four women withdrew, choosing to follow up at alternate facilities. As such, 96 of the 120 women who participated in the early prenatal phase had assessments of depression and immune profiles during the late pregnancy phase. Pregnant women with axis-I mental disorders other than depression—such as bipolar disorder, schizophrenia, autism, substance abuse disorders, psycho-organic disorders, and generalized anxiety disorder—were excluded from the study. We also excluded women with immune, autoimmune, and neuroinflammatory disorders, including inflammatory bowel disease, multiple sclerosis, type 1 diabetes mellitus, systemic lupus erythematosus, rheumatoid arthritis, and psoriasis, as well as women who were taking antidepressants, mood stabilizers, antipsychotic drugs, or immunosuppressive drugs. During the course of the investigation, only supportive psychotherapy, psychoeducation, sleep hygiene education, and family counseling were permitted. Antidepressants, antioxidant supplements, or other medical treatments for depression have not been administered to any of the pregnant women.

Institutional Review Board (IRB) permission was received from both Khon Kaen Hospital and Chulalongkorn University (KEF62036 and COA No. 280/2019). All women provided their informed consent in writing.

### 2.2. Measurements

The sociodemographic data included age, marital status, family income, and the age of the spouse. The women’s medical records were examined for their gravidity, parity, pregnancies, body mass index (BMI), and perinatal depression history. The EPDS was used to evaluate depressed symptoms and the degree of depression [28]. The scale consists of 10 items scored from 0 to 3 on a four-point Likert scale, with a total score ranging from 0 to 30. This scale measures the emotional state of pregnant women throughout the previous week. In the Thai version of the EPDS, the validated screening cutoff value for pregnant women was ≥10 (AUC 0.84, sensitivity 60%, and specificity 90%) [29]. This scale has strong reliability and validity, as indicated by research conducted in the upper Northeast area of Thailand among pregnant women in their third trimester [30]. Relationship satisfaction was evaluated using a four-point Likert scale ranging from “satisfied” to “dissatisfied”. In addition, we evaluated if the pregnancy was unintended (dummy variable: yes/no). The Abuse Assessment Screen for intimate partner violence (AAS) [31,32] was used to assess partner abuse during pregnancy among our pregnant women. We used item one (AAS1) in the current study since it was the most important in a previous study [7]. The Multidimensional Scale of Perceived Social Support (MSPSS) [33], a 12-item rating measure scored on a seven-point Likert scale (1 to 7 for each item), was used to assess perceived social support [33]. The MSPSS friends subdomain was chosen as the most important domain for this study [7]. The Adverse Childhood Experiences (ACE) Questionnaire was used to evaluate adverse childhood experiences [34]. In the current study, we used [7] psychological abuse, physical abuse, sexual abuse, a formative construct made using the three abuse scores (labeled ACE_abuse), any neglect (mental or physical neglect), a PC extracted from psychological and physical abuse, domestic violence, a family history of substance abuse, and a family history of psychiatric illness (labeled ACE_FDA, ACE-Family-Dysfunction and Abuse [7].

To measure PMS symptoms, the Premenstrual Symptoms Screening Test (PSST) was used. The premenstrual symptoms (items 1–14) and their repercussions in different contexts (A–E: five items) were assessed on a four-point Likert scale ranging from “not at all” to “mild” to “moderate” to “severe” and scored on a scale from 1 to 4 [35]. In the current study, we used the first PC extracted from ten PSST reflecting the affective component of PMS (labelled PSST_depression), the first PC extracted from three physiosomatic PSST symptoms (labelled PSST_physiosom), the first PC extracted from all five PSS interference data (A, B, C, D, and E) (labelled PSST_inference), and the total PSST score [12].

### 2.3. Assays

BD Vacutainer^®^ EDTA (10 mL) tubes were used to collect fasting blood in the early morning hours. Blood was sampled from 120 women in the early pregnancy phase (<16 gestational weeks) and in 96 of those women in the late pregnancy phase (>24 gestational weeks). We determined the concentrations of cytokines, chemokines, and growth factors in plasma samples using the Bio-Plex Pro human cytokine 27-plex test kit (BioRad, Carlsbad, California, United States of America). The LUMINEX 200 apparatus was used to measure the fluorescence intensities (FI) of the detecting antibodies and streptavidin-PE (BioRad, Carlsbad, California, United States of America). The intra-assay coefficient of variance of all analytes were less than 11 percent. Appendix A lists the names, acronyms, and official gene symbols for all the cytokines/chemokines/growth factors measured in this study. In the current investigation, we chose to perform statistical analyses on the FI data (with the blank analyte removed) since FI values are often a superior alternative to absolute concentrations, particularly when more than one plate is used [21,22]. IL-1β, IL-2, IL-5, IL-7, IL-10, IFN-γ, CCL5, and VEGF exhibited a high number of results below the assay’s sensitivity (>60%) and, therefore, were excluded from the analysis performed on the single indicators. Nevertheless, these analytes were used to compute the composite scores when at least 7% (n = 15) of the assays were measurable (thus all, except IL-5). As such, eight analytes were excluded from the single indicator analyses, but IL-1β, IL-2, IL-7, IL-10, IFN-γ, CCL5, and VEGF could be used as prevalences in the computation of composite scores. ESF, Appendix A provides a summary of the immunological profiles studied in this investigation [20,21,22]. Where needed, cytokines were used after logarithmic (log10), square root, fractional rank-based normal transformations, or Winsorization, and all data were processed as z transformations.

## 3. Data Analysis

We used analysis of variance (ANOVA) to compare scale variables across research groups and Chi-square (χ^2^) or Fisher’s exact probability test to compare categories. Using Pearson’s product-moment correlation coefficients and the point-biserial correlation coefficient, correlations between scale variables or scale and binary variables were analyzed. Using Generalized Linear Model (GLM) analysis, the connection between the EPDS score and immune profiles was investigated. An automatic stepwise binary logistic regression analysis was performed to determine the most accurate predictors of prenatal depression versus the absence of antenatal depression using the immune profiles, PSST and ACE scores, relationship satisfaction, planned versus unplanned pregnancy, AAS1, and the MSPSS_friends score. Using multiple regression analysis, the most influential factors (immune and psychological variables and PMS) affecting the EPDS score were determined. In addition to the manual regression technique, we used an automated technique with *p*-values of 0.05 for model entrance and 0.10 for model exclusion. We determined the model statistics (F, df, and *p* values) and the total variance explained (R^2^), as well as the standardized β coefficients with t statistics and *p*-values for each predictor. Additionally, the variance inflation factor and tolerance were evaluated to identify any collinearity or multicollinearity concerns. Heteroskedasticity was determined with the use of the White and modified Breusch-Pagan homoscedasticity tests. Generalized estimating equations (GEE), repeated measures, were employed to examine the associations between the repeated assessments of the EPDS and biomarker data. For statistical significance, two-tailed tests with a *p*-value of 0.05 were utilized. The various immune profiles were investigated using principal component analysis (PCA) to check whether one general immune PC could be extracted. The KMO statistic (cut-off value > 0.6), Bartlett’s test, and the anti-image correlation matrix were used to evaluate the data’s factoriability. To be accepted as a valid PC, the variance explained should be >50.0% and all loadings should be >0.7 on this first PC. We used two-step cluster analysis to delineate a cluster of patients with increased EPDS scores and immune profiles, both entered as continuous variables, and the antenatal depression diagnosis entered as a categorical variable. The number of clusters was determined automatically, and the cluster solution was evaluated using the silhouette measure of cohesion and separation (threshold value > 0.5). All statistical analyses were conducted using version 28 of IBM, SPSS for Windows.

## 4. Results

Results of cluster analyses performed on the early phase data.

We found that the early phase EPDS score was strongly associated with the total PSST score (r = 0.688, *p* < 0.001, n = 120), relation dissatisfaction (r = 0.406, *p* < 0.001), unplanned pregnancy (r = 0.413, *p* < 0.001), and AAS1 (r = 0.403, *p* < 0.001). A large part (31.0%; F = 13.34, df = 3/111, *p* < 0001) of the variance in the total EPDS score was explained by the combined effects of unplanned pregnancy (*p* < 0.001), relation dissatisfaction (*p* = 0.017), and AAS1 (*p* < 0.001). Since these psychological factors explain such a large part of the variance, we also computed residualized EPDS (resEPDS) values after covarying for relation dissatisfaction, unplanned pregnancy, and AAS1. This new EPDS score thus reflects severity of depression independent of the effects of adverse psychological stressors.

A two-step cluster analysis was performed with the diagnosis of antenatal depression (EDPS ≥ 10) as categorical factor, total EPDS and resEPDS scores, and the three major cytokine profiles (M1, Th-1, and Th-17) as continuous variables. A two-cluster solution yielded the best solution with a silhouette measure of cohesion and separation of 0.68, whereby the first cluster comprised 76 people and the second cluster comprised 44 people. Figure 1 shows that both EPDS scores and all immune profiles, except CIRS, were significantly higher in cluster two than in cluster one. ESF, Appendix A shows the results of univariate GLM analyses of the immune profiles with age and BMI as covariates. The top three most important profiles of cluster two were (in descending order): Th-1, IRS, and T cell growth. ESF, Appendix A shows the outcome of univariate GLM analyses performed on the single cytokines-growth factors. IL4, IL-6, IL-8, IL-12_p70_, IL-15, IL-17, and GM-CSF were significantly higher in cluster two than in cluster one. ESF, Appendix A shows a clustered bar graph with the differences in those cytokines.

### 4.1. Socio-Demographic and Clinical Data

Table 1 shows the socio-demographic and clinical data in both clusters. The range of the early EPDS score was 0–20, and the range of the late EPDS score was 0–16. There were no significant differences in age, pre-pregnancy BMI, gravidity, parity, and a history of Caesarian delivery between both clusters. Unplanned pregnancy, relation dissatisfaction, and AAS1 were all significantly higher in cluster two than in cluster one. PSST_depression, PSST_physiosom, PSST_interference, and PSST_total were significantly higher in cluster two than in cluster one, whilst MSPSS_friends was significantly lower in cluster two. Psychological abuse, physical abuse, total abuse, ACE1, domestic violence, and family dysfunction were significantly higher in cluster two than in cluster one. Binary logistic regression analysis showed that PSST_interference (Wald = 6.02, df = 1, *p* = 0.014), PSST_depression (Wald = 5.30, *p* = 0.005), and All_abuse (Wald = 9.17, *p* = 0.002) predict cluster two versus cluster one (χ2 = 43.57, df = 3, *p* < 0.001) with a Nagelkerke value of 0.416. The total number of correctly classified patients was 84.2% (sensitivity = 77.3% and specificity = 88.2%). As such, the cluster analysis performed on the early pregnancy data has generated a cluster of pregnant women with increased adverse experiences, immune, and depression scores.

### 4.2. Prediction of the Early EPDS Scores Using Immune Profiles

Multiple regression analysis showed that 67.5% of the variance in the early phase EPDS score (F = 36.50, df = 5/88, *p* < 0.001) was explained by the regression on PSST_total (β = 0.496, t = 7.58, *p* < 0.001), ACE1 (β = 0.221, t = 3.43 *p* < 0.001), MSPSS_friends (β = −0.257, t = −4.06, *p* < 0.001), relationship dissatisfaction (β = 0.182, t = 2.80, *p* = 0.006) and AAS1 (β = 0.171, t = 2.62, *p* = 0.010). Table 2 shows that the early phase M1, Th-1, Th-2, Th-17, IRS, and growth factor profiles, as well as the IRS/CIRS ratio, had significant effects on the early phase EPDS score above and beyond the effects of the PSST_total, MSPSS_friends, ACE1, AAS1, and relationship dissatisfaction scores (entered as five covariates, forced entry). Figure 2 and Figure 3 show the partial regressions of the baseline EPDS score on the early phase Th-1 and Th-17 profiles, respectively. Of the seven single cytokines that were increased in cluster two, we found that CXCL8 (IL-8), IL-15, IL-17, and GM-CSF were significantly associated with the early EPDS score.

### 4.3. EPDS and Immune Profiles in the Late Pregnancy Phase

We examined whether the late phase EPDS score could be explained by independent effects of immune profiles. A first multiple regression analysis (F = 12.14, df = 4/86, *p* < 0.001) showed that 36.1% of the variance in the late phase EPDS score was explained by the early phase EPDS score (β = 303, t = 3.36, *p* = 0.001), mental neglect (β = 0.261, t = 2.96, *p* = 0.004), AAS1 (β = 0.253, t = 2.85, *p* = 0.005), and late phase Th-2 (β = 0.234, t = 2.71, *p* = 0.008). In addition, the late phase Th-17 (β = 0.189, t = 2.15, *p* = 0.035) and CIRS (β = 0.235, t = 2.68, *p* = 0.009) profiles had significant effects on late phase EPDS, above and beyond the effects of the early EPDS score (which is affected by immune profiles), mental neglect, and AAS1.

ESF, Appendix A shows the results of GEE (repeated measurements) analyses with the early and late EPDS and immune profiles as dependent variables, and time (early-late pregnancy phase), the early pregnancy clusters, and their interaction as explanatory variables. There was a significant effect of the interaction pattern time × diagnosis (and significant time and diagnosis effects) on the repeated EPDS assessments. We found a significant increase in cluster one people (*p* = 0.010), but in cluster two there was a significant decrease in the EPDS score from the early to the late phase (*p* < 0.001). The same table shows the increased late EPDS values in cluster one are accompanied by increased Th-1, Th-17, IRS, growth factor, and T cell growth profiles. The lowered late EPDS scores in cluster two are accompanied by lowered Th-1 and Th-17 profiles.

Using GEE analyses (repeated measurements), we found that the EPDS scores (the early to the late phase) were significantly and positively associated with the Th-1 (Wald = 4.73, df = 1, *p* = 0.030), Th-2 (Wald = 4.02, df = 1, *p* = 0.045), Th-17 (Wald = 7.32, df = 1, *p* = 0.007), IRS (Wald = 4.71, df = 1, *p* = 0.030), and CIRS (Wald = 5.55, df = 1, *p* = 0.018) profiles. Of the seven cytokines that were increased in cluster two, we found that the repeated measurements of IL-4 (Wald = 5.50, df = 1, *p* = 0.019), IL-6 (Wald = 4.21, df = 1, *p* = 0.040), IL-15 (Wald = 4.20, df = 1, *p* = 0.040), and IL-17 (Wald = 5.65, df = 1, *p* = 0.017) were significantly associated with those of the EPDS score.

## 5. Discussion

### 5.1. Immune Profiles of Early Prenatal Depression

The first major findings of this study are that (a) all immune profiles (except CIRS) during early pregnancy were significantly linked with the EPDS score, independent of the effects of psychological variables and PMS; and (b) elevated CXCL8 (IL-8), IL-15, IL-17 and GM-CSF are the primary cytokines associated with prenatal depression, while IL-4, IL-6 and IL-12 associate with an increased EPDS score in cluster 2 participants. Previous research has shown that prenatal depression is accompanied by elevated C-reactive protein, a positive acute phase protein, and decreased zinc, a negative acute phase reactant, suggesting a modest inflammatory response [1,18]. Our findings that IL-6 is linked with the severity of prenatal depression are consistent with those of previous studies [19,36,37]. These authors also found that IL-1β [19] and TNF-α [34] may be related with moderate depression symptoms in early pregnancy to midgestation. In contrast to the findings of Sha et al. [19], we discovered that CXCL8 (IL-8) was associated with the severity of the condition. Increased CXCL8 was also detected by Corwin et al. [38] who reported that postpartum depression might be predicted by a high IL-8/IL-10 ratio. Karlsson et al. [39] found that although IL-6 and TNF-α were unrelated to prenatal symptoms of depression, IL-12, IL-9 and IL-13, as well as the IFN-γ/IL-4 ratio and IL-5, were all positively associated with depressive symptoms.

In addition, Maes et al. [40] revealed that pregnant women with a history of major depression had elevated levels of IL-6 and soluble IL-1 receptor antagonist (sIL-1RA), suggesting sensitization of the immune-inflammatory response. In accordance, Christian et al. [41] observed that blood levels of macrophage migration inhibitory factor (MIF) were higher after one week of influenza vaccination in pregnant women with greater depressive ratings, suggesting immunological sensitization. Research has connected prenatal depression to inflammatory pregnancy morbidities such as preeclampsia, preterm birth, and gestational diabetes [42,43,44,45,46].

Current hypotheses and reviews demonstrate that immunological pathways that contribute to non-pregnancy major depression also contribute to perinatal depression. Hence, Leff-Gelman et al. [47] postulated that the same immune-inflammatory pathways linked with major depression also operate in the placenta during prenatal depression. Activation of Toll-Like Receptors (TLR) in placental immune cells, for instance, may promote a Th1/Th17 shift with an increase in pro-inflammatory cytokines that cause depressive symptoms [47]. Our analysis demonstrates, however, that prenatal depression is accompanied by hyperactive Th-1 as well as Th-17 phenotypes. The comprehensive review of Roomruangwong et al. [1] reported that not all neuro-immune and neuro-oxidative pathways that define non-pregnancy-related depression have a role in perinatal depression, including increased bacterial translocation. The pregnancy-specific effects of sex hormones (oestrogen and progesterone) and the significant activation of the hypothalamic-pituitary-adrenal (HPA) axis during pregnancy may explain some of the discrepancies in the literature, as explained below.

### 5.2. IRS and CIRS Activity throughout the First Trimester

The second major finding is that, as with non-pregnancy-related MDD [20,21,22], prenatal depression severity is linked with greater IRS vs. CIRS responses, despite the presence of a robust Th-2 response and elevated IL-4 levels. The latter may impose negative feedback on M1 and Th-1 phenotypes [20]. Moreover, elevated glucocorticoid levels during pregnancy may attenuate the synthesis and release of Th-1 and M1 cytokines, encouraging a Th-2 shift [48]. Thus, the observed levels of IRS (M1, Th-1, and Th-17) and CIRS (Th-2 and Treg) cytokines throughout various phases of pregnancy are dictated by the tight balance between IRS and CIRS activity and elevated glucocorticoid levels. In the past, we have hypothesized that enhanced HPA-axis activity during pregnancy may reduce IRS activation during pregnancy, while boosting Th-2 and Treg activity [1]. Complicating matters, there is a negative correlation between inflammatory markers and cortisone levels in healthy pregnant women, but not in those with prenatal depression [49,50], whereas other authors hypothesized that dysfunctions of the HPA-axis may be mediated by immunological and placental mechanisms [51]. Pregnancy-specific alterations in sex hormones, particularly progesterone, may further modulate the homeostatic setpoint between the IRS and CIRS phenotypes [1].

These hormonal changes may explain why the IRS pathway alterations seen in this study are substantially less profound than those observed in non-pregnant MDD/BD patients [1,20,22]. These hormonal effects may also explain why not all research on pro-inflammatory characteristics in prenatal or perinatal depression had positive outcomes. For example, a first investigation was unable to find a link between IL-6, soluble IL-6 receptor (sIL-6R), and sgp130 (thus measuring IL-6 signaling, including trans-signaling), sIL-1RA, and leukaemia inhibitory factor and postpartum depression, despite substantial correlations with postpartum blues [52]. A pilot study reported an inverse connection between IL-1β, TNF-α, and IL-17 and perinatal depression [53]. TRAIL (TNF-α-related apoptosis inducing ligand), M-CSF (macrophage colony-stimulating factor), and CX3CL1 (fractalkine) were reduced in prenatal depression, according to Edvinsson et al. [54].

### 5.3. Variations in Immune Profiles between Early and Late Pregnancy

The third major finding of this study is that there was a shift in immune profiles from early to late pregnancy, characterized by a sharp increase in the IRS/CIRS ratio due to a decrease in CIRS and increases in the M1 and growth factor profiles, indicating that the other profiles (Th-1, Th-2, Th-17, and CIRS) remained more stable over time. Earlier, Blackmore et al. [55] demonstrated that cytokines, particularly IL-6 and TNF-α, are remarkably stable in pregnant women. However, the actual changes in EPDS from early to late pregnancy were positively associated with Th-2 and Th-17 phenotypes, indicating that again, the balance between different types of IRS and CIRS (as well as glucocorticoids and sex hormones) impact the severity of late prenatal depression, and consequently postnatal depression, which is strongly predicted by prenatal depression [1].

Pregnancy is known to cause alterations in the immune system’s equilibrium. The invasion of the uterine epithelium by the blastocyst during implantation triggers an immune-inflammatory response [56,57,58], in which endometrial cells and immune cells alike produce increased quantities of IL-6, TNF-α, and CXCL8 [59]. In the second phase, the immune system gains a more anti-inflammatory mode [58], as seen by enhanced CIRS responses in our research. Induction of an inflammatory response in the latter days of pregnancy facilitates uterine contractions, birth, placental rejection, and protection against puerperal sepsis [58,60]. Hence, symptoms of enhanced peripheral inflammation are present towards the end of a full-term pregnancy but are kept in check by heightened immunoregulatory feedback systems, which are achieved at least in part by higher HPA-axis activity and progesterone levels, as well as immunological homeostatic mechanisms [1]. As such, the immune pathophysiology of prenatal depression is not comparable with that of non-pregnancy-associated depression.

### 5.4. Results of Cluster Analysis

The fourth major finding of this study is that we were able to identify a cluster of early pregnancy women with elevated EPDS scores, MCAS/PMS, immunological profiles, adverse childhood experiences, and elevated current psychological stressors, such as unexpected pregnancy and intimate partner dissatisfaction. As a result, these early and current adverse life experiences, PMS, depressive symptoms, and IRS activation shape a cluster of pregnant women that is well-defined. Notably, the latter class is statistically more significant than the diagnosis of prenatal depression as determined by an EPDS ≥ 10 cutoff score. Moreover, this new phenotypic class was also characterized by increased levels of pro-inflammatory (IL-6, IL-8, IL-12, IL-15, IL-17, and GM-CSF) and anti-inflammatory (IL-4) cytokines. Our results suggest that the combined effects of these psychological (ACE and acute psychological stressors) and biological (immune profiles and PMS) factors, plus the ensuing early pregnancy-related depressive symptoms, form a new phenotypic class. The latter combines a lifetime trajectory of psychological (ACE) and hormonal stressors (PMS) and neuro-immune pathways (PMS) with the ensuing prenatal depressive symptoms and activated immune networks.

In the current investigation, however, we determined that these factors yielded cumulative effects and that there were no significant connections between ACE or PMS and the immunological profiles of prenatal depression. In contrast to earlier research, which indicated that ACE, PMS or menstrual distress, and current psychological stressors impact indicators of immunological IRS activation, the current study revealed no such association. First, ACE were shown to be highly related with the stimulated production of cytokines, chemokines, and growth factors by stimulated whole blood culture supernatant, but not unstimulated whole blood culture supernatant, suggesting immunological sensitization to immune injury [21,22]. In the current investigation, however, we examined plasma levels of the immunological products, which are more closely related to unstimulated levels in culture supernatant than stimulated levels. Second, PMS and menstrual distress are related to elevated levels of proinflammatory chemokines, acute phase reactants, and tryptophan catabolites, which are triggered by inflammatory processes [13,25,61]. Third, ACEs and psychosocial stresses in adult life are known to stimulate immune-inflammatory cytokines including IL-6 [21,26,27]. Even modest psychological stressors, such as examination stress, may stimulate the production of pro-inflammatory cytokines, including IL-6, TNF-α and IFN-γ [62]. Nevertheless, such changes in the prenatal period are likely obscured by the impact of the greatly elevated HPA-axis activity and progesterone levels, which alter the homeostatic equilibrium between IRS and CIRS phenotypes from early to late pregnancy [1]. Our results that psychological stressors and PMS had less of an influence on the late EPDS score than the early EPDS score are similarly suggestive of such confounding effects.

### 5.5. Limitations

This work would have been more intriguing if we had assessed HPA-axis activity and plasma progesterone levels, as well as cytokines, chemokines, and growth factors, in the supernatant of stimulated whole blood cultures.

## 6. Conclusions

Independent of the impact of psychosocial factors and PMS, all immunological profiles, including M1, Th-1, Th-2, and IRS, were significantly correlated with the early EPDS score. Cluster analysis revealed that the combined impacts of ACE, relationship dissatisfaction, undesired pregnancy, PMS, and activated immunological profiles (including elevated levels of IL-4, IL-6, CXCL8, IL-12, IL-15, IL-17, and GM-CSF), as well as the resulting early depressive symptoms, define a new endophenotypic class. Immune profiles changed from early to late pregnancy, with a significant rise in the IRS/CIRS ratio. Immune profiles, specifically the Th-2 and Th-17 phenotypes, predicted the actual changes in EPDS score from early to late pregnancy. Activation of the immune system contributes to depressive symptoms during pregnancy.

## Figures and Tables

**Figure 1 brainsci-13-00727-f001:**
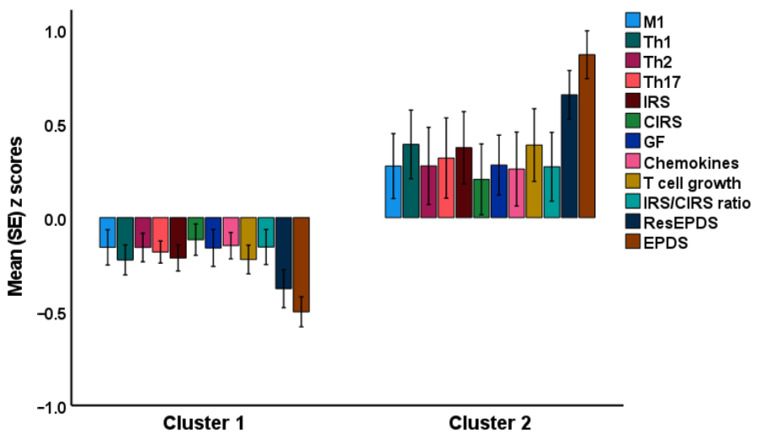
Results of cluster analysis performed on EPDS (Edinburgh Postnatal Depression Scale) scores and immune profiles. M1: macrophage M1, Th: T helper, IRS: immune-inflammatory response system, CIRS: compensatory immunoregulatory system, GF: growth factors, ResEPDS (residualized EPDS after covarying for psychological stressors). All measurements (except CIRS) are significantly higher in cluster two than in cluster one.

**Figure 2 brainsci-13-00727-f002:**
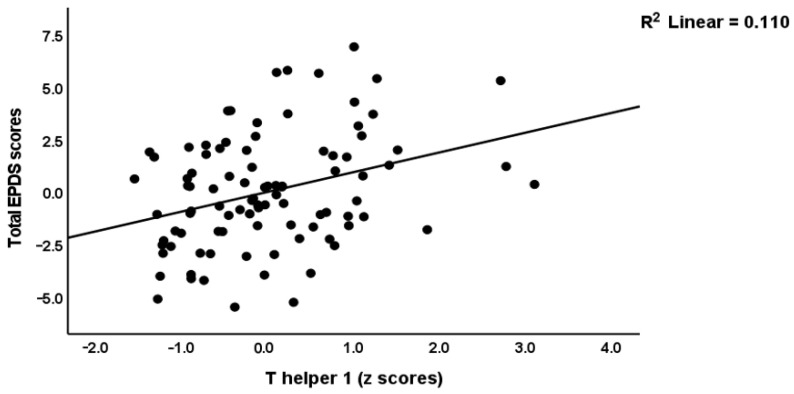
Partial regression of the early EPDS (Edinburgh Postnatal Depression Scale) score on the T helper 1 profile.

**Figure 3 brainsci-13-00727-f003:**
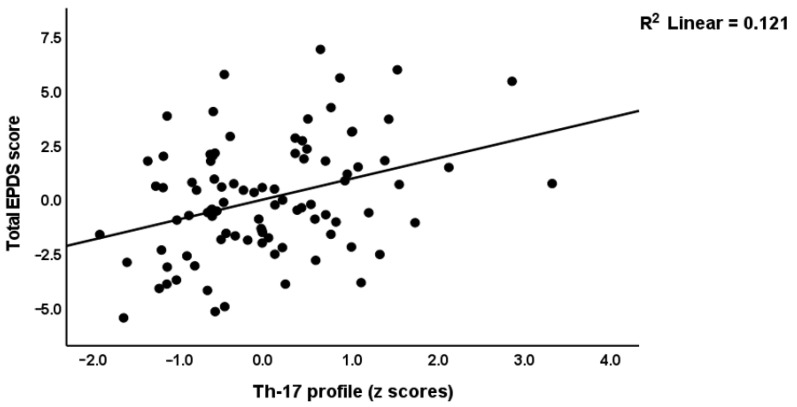
Partial regression of the early EPDS (Edinburgh Postnatal Depression Scale) score on the T helper 17 profile.

**Table 1 brainsci-13-00727-t001:** Socio-demographic and clinical data of the cluster analysis-generated classes.

Variables	Cluster 1 n = 76	Cluster 2 n = 44	F, χ2	df	*p*
Early EPDS	3.42 (2.55)	10.41 (4.53)	117.45	1/118	<0.001
Residualized early EPDS	−0.378 (0.879)	0.653 (0.853)	39.20	1/118	<0.001
Late EPDS	4.34 (2.78)	6.29 (4.15)	7.42	1/118	0.008
Age (years)	27.0 (5.2)	27.5 (6.1)	0.23	1/118	0.635
Gravidity (1/2/≥3)	25/35/16	19/13/12	3.17	2	0.205
Parity (0/1/≥2)	28/42/6	21/17/6	3.29	2	0.193
History of Caesarian delivery (N/Y)	64/12	37/7	0.00	1	1.0
Basal body mass index (kg/m^2^)	23.2 (5.6)	23.6 (4.7)	0.16	1/118	0.689
Premenstrual syndrome (N/Y)	70/6	31/13	9.80	1	0.002
PSST_depression	−0.359 (0.702)	0.619 (1.134)	34.05	1/118	<0.001
PSST_physiosom	−0.172 (0.951)	0.297 (1.022)	6.42	1/118	0.013
PSST_interference	−0.352 (0.740)	0.607 (1.103)	32.40	1/118	<0.001
PSST_total	24.21 (4.95)	30.43 (7.38)	30.41	1/118	<0.001
Planned pregnancy (N/Y)	19/57	25/19	12.15	1	<0.001
Intimate relation satisfaction (N/Y)	0/76	10/34	18.84	1	<0.001
AAS1 (N/Y)	69/7	33/11	5.45	1	0.020
ACE psychological abuse	2.33 (0.87)	2.93 (1.55)	7.49	1/118	0.007
ACE physical abuse	2.03 (0.16)	2.43 (1.26)	7.64	1/118	0.007
ACE sexual abuse	0.05 (0.36)	1.37 (0.41)	1.36	1/118	0.246
ACE All_abuse	4.40 (0.94)	5.50 (2.80)	9.72	1/118	0.002
ACE Any neglect	17.07 (6.31)	19.36 (7.73)	3.13	1/118	0.080
ACE_Family dysfunction abuse	8.50 (1.18)	10.05 (4.30)	8.72	1/118	0.004
ACE1	1.24 (0.61)	11.66 (0.99)	8.42	1/118	0.004
ACE_total	26.01 (6.89)	30.05 (10.50)	6.44	1/118	0.012
MSPSS_friends	5.54 (1.06)	4.98 (1.27)	6.62	1/118	0.011

All results are shown as mean (SD) or as frequencies. F/χ2 all results of analysis of variance (ANOVA), and analysis of contingency tables (χ2-Test). EPDS: Edinburgh Postnatal Depression Scale, total score; Residualized EPDS: after covarying for the effects of psychological stressors; PSST_: principal components extracted from the Premenstrual Symptoms Screening Test (PSST) score (depression, physiosomatic, or interference domains); AAS1: Abuse Assessment Screen, first item (experience of emotional or physical abuse); ACE1: Adverse Childhood Experiences, first item (psychological abuse); ACE_FDA: Family Dysfunction Abuse; MSPSS: Multidimensional Scale of Perceived Social Support, friends subdomain.

**Table 2 brainsci-13-00727-t002:** Results of multiple regression analysis with the early pregnancy Edinburgh Postnatal Depression Scale (EPDS) score as dependent variable and the early immune profiles or single cytokine values as explanatory variables.

Profile	B	SE	β	t	*p*
M1	0.716	0.290	0.159	2.47	0.016
Th-1	0.968	0.291	0.214	3.33	0.001
Th-2	0.722	0.302	0.157	2.39	0.019
Th-17	0.934	0.277	0.202	3.73	0.001
IRS	0.964	0.290	0.211	3.32	0.001
CIRS	0.564	0.287	0.132	1.97	0.052
GF	0.609	0.298	0.129	2.04	0.044
Chemokines	0.431	0.306	0.092	1.41	0.193
IRS/CIRS	0.684	0.300	0.149	2.28	0.025
CXCL8 or IL-8	0.145	0.069	0.131	2.10	0.038
IL-15	0.882	0.276	0.191	3.19	0.002
IL-17	0.907	0.267	0.211	3.40	0.001
GM-CSF	0.148	0.067	0.134	2.21	0.029

M1: macrophage M1, Th: T helper, IRS: immune-inflammatory response system, CIRS: compensatory immunoregulatory system, GF: growth factors, IL: interleukin, GM-CSF: granulocyte-macrophage colony stimulating factor.

## Data Availability

The dataset generated during and/or analyzed during the current study will be available from the corresponding author (MM) upon reasonable request and once the dataset has been fully exploited by the authors.

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
