# Peer review of "The Cytokine, Chemokine, and Growth Factor Network of Prenatal Depression"

_brainsci, 2023, doi:10.3390/brainsci13050727_

Round 1

Reviewer 1 Report

The article addresses an important interdisciplinary issue: the relationship between immunological markers and pregnancy-related depression.

The abstract is constructed correctly, accurately reflecting the essence of the presented article.

The introduction provides a comprehensive review of the research on the topic, but it is necessary to go a bit more controversial and consider the relationship between inflammation and psychological stress. Do individuals prone to psychotraumatic circumstances not also have an underlying increase in inflammatory markers?

Materials and methods are explained and presented in detail. The results are visualized and presented in detail and clearly.

The discussion is coherently presented and reflects the comparison with the other studies cited. Some extension may be considered to match the above changes to the introduction.

The conclusion is clear and well presented.

The reviewer

The article is written in fluently formulated and stylistically well-structured English. No linguistic correction needed.

The Reviewer

Author Response

The article addresses an important interdisciplinary issue: the relationship between immunological markers and pregnancy-related depression.

The abstract is constructed correctly, accurately reflecting the essence of the presented article.

The introduction provides a comprehensive review of the research on the topic, but it is necessary to go a bit more controversial and consider the relationship between inflammation and psychological stress. Do individuals prone to psychotraumatic circumstances not also have an underlying increase in inflammatory markers?

@@ANSSWER: addressed in the intro as:

In addition, in humans, psychosocial stressors may induce mild immune-inflammatory responses [26,27].

Materials and methods are explained and presented in detail. The results are visualized and presented in detail and clearly.

The discussion is coherently presented and reflects the comparison with the other studies cited. Some extension may be considered to match the above changes to the introduction.

@@ANSWER: addressed in the discussion as:

ACEs and psychosocial stresses in adult life are known to stimulate immune-inflammatory cytokines including IL-6 [21, 26-27].

The conclusion is clear and well presented.

The reviewer

Reviewer 2 Report

The review of the manuscript entitled: “The cytokine, chemokine, and growth factor network of prenatal depression”. In this study the Authors aimed to investigate the influence of immune profiles on severity of prenatal depression.

Comments for Authors:

Thank you for the valuable research you have done. The study is well conducted and concerns interesting topic and acceptable writing. However, there are some questions:

1) ‘Methods’ section, ‘Assays’ subsection: The authors mentioned “Blood was sampled from 120 women in the early pregnancy phase (< 16 gestational weeks) and in 96 of those women in the late pregnancy phase (> 24 gestational weeks)”. Why the 24 women who withdrew, did not excluded from data and analysis?

2) What was the overall range of EPDS scores in participants?

3) Did participants receive any kind of treatment for depression during study period?

Good luck

The study's writing is acceptable.

Author Response

The review of the manuscript entitled: “The cytokine, chemokine, and growth factor network of prenatal depression”. In this study the Authors aimed to investigate the influence of immune profiles on severity of prenatal depression.

Comments for Authors:

Thank you for the valuable research you have done. The study is well conducted and concerns interesting topic and acceptable writing. However, there are some questions:

1) ‘Methods’ section, ‘Assays’ subsection: The authors mentioned “Blood was sampled from 120 women in the early pregnancy phase (< 16 gestational weeks) and in 96 of those women in the late pregnancy phase (> 24 gestational weeks)”. Why the 24 women who withdrew, did not excluded from data and analysis?

@@ANSWER: why should we delete 24 subjects to assess the baseline values and loose some df and statistical power?

2) What was the overall range of EPDS scores in participants?

@@ANSWER: this is now addressed in the results section as:

The range of the early EPDS score was 0-20, and that of the late EPDS 0-16.

3) Did participants receive any kind of treatment for depression during study period?

@@ANSWER: this is now addressed in the text as:

During the course of the investigation, only supportive psychotherapy, psychoeducation, sleep hygiene education, and family counseling were permitted. Antidepressants, antioxidant supplements, or other medical treatments for depression have not been administered to any of the pregnant women.